# Vulvar Lichen Planus Presenting as Chronic Vulvar Purpura

**DOI:** 10.3390/medicina59020206

**Published:** 2023-01-20

**Authors:** Boštjan Luzar, Anamarija Truden, Lara Turk, Vid Janša, Fiona Lewis, Eduardo Calonje, Špela Smrkolj

**Affiliations:** 1Institute of Pathology, Faculty of Medicine, University of Ljubljana, 1000 Ljubljana, Slovenia; 2Department of Gynaecology and Obstetrics, Faculty of Medicine, University of Ljubljana, 1000 Ljubljana, Slovenia; 3Division of Gynaecology and Obstetrics, Department of Gynaecology, University Medical Centre, 1000 Ljubljana, Slovenia; 4St John’s Institute of Dermatology, Guy’s & St Thomas NHS Trust, London SE1 7EH, UK; 5Department of Dermatopathology, St John’s Institute of Dermatology, St Thomas’s Hospital, London SE1 7EH, UK

**Keywords:** chronic vulvar purpura, vulvar lichen planus, Zoon’s vulvitis, lichen aureus, pigmented purpuric dermatosis

## Abstract

*Background*: There is a broad spectrum of vulvar pigmented lesions that differ based on their histopathological and clinical features. Chronic vulvar purpura is a rare entity, associated with a broad morphological spectrum, from lichen aureus, Zoon’s vulvitis, pigmented purpuric dermatosis and with lichen planus as in our case. *Case presentation:* In this article we discuss a case of an 86-year-old white woman with hyperpigmentation on her upper vulva, next to the introitus, with complaints of urine incontinence. Biopsy revealed subepithelial stromal lichenoid inflammatory infiltrate containing plasma cells, lymphocytes and some neutrophilic granulocytes as well as dilated and congested vessels. Hemosiderin deposits and erythrocyte extravasation were found. There was evidence of hyperkeratosis with hyper granulosis and erosions. Spongiosis was also noted. Few melanocytes were identified with no sign of malignancy. These findings correlate with the diagnosis of vulvar lichen planus. *Conclusions:* Chronic vulvar purpura is a clinical term used for different chronic inflammatory dermatoses presenting as red bluish or violaceous discolorations on the vulva, often associated with cayenne-pepper-like speckling. Considering a great overlap of possible diseases, the final diagnosis could be challenging. It is important to exclude a melanocytic tumour in these cases.

## 1. Introduction

Chronic vulvar purpura is a clinical term used for different chronic inflammatory dermatoses presenting as red, bluish or violaceous discoloration on the vulva. The most common diseases associated with chronic vulvar purpura include Zoon’s vulvitis, lichen aureus, pigmented purpuric dermatosis and vulvar lichen planus [1].

Morphological changes in Zoon’s vulvitis have generally been attributed to a combination of irritation/moisture and mechanical/friction trauma and are as such non-specific. Zoon’s vulvitis can either develop as an isolated condition or represents an unrelated phenomenon in the background of a more specific inflammatory process. Clinical and morphological overlap between vulvar pigmented purpuric dermatosis and Zoon vulvitis suggests a site-specific mucosal reaction to an erosive process, either inflammatory (hypersensitivity reaction) or traumatic. Genital lichen planus can present clinically as chronic vulvar purpura, but is distinguished from potential mimickers by the presence of interface/lichenoid tissue reaction associated with vacuolar degeneration of basal keratinocytes and necrotic basal keratinocytes [1,2].

Pigmented purpuric dermatoses presenting at genital sites are exceedingly rare, with only a few cases reported in the literature [1,3]. No medical intervention is of consistent benefit for the treatment of the pigmented purpuric dermatoses. Pruritus may be alleviated using topical corticosteroids and antihistamines. Associated venous stasis should be treated by compression hosiery [1].

We are reporting a case of an 86-year-old woman presented with a hyperpigmented patch and papule on the vulva.

## 2. Case Presentation

### 2.1. Clinical Findings

An 86-year-old Caucasian woman was admitted to the Division of Gynaecology and Obstetrics, University Medical Centre in Ljubljana, Slovenia due to darkly pigmented irregular patch on her vulva, of six-month duration. She did not complain about the presence of flares of pruritus, burning or pain, which are otherwise typical symptoms of vulvar lichen planus. Physical examination revealed a well-demarcated brownish to blackish macule with irregular borders measuring 3 × 2 cm in diameter located at the vestibule and over the clitoral frenulum (Figure 1, asterix). In addition, a satellite papule with indistinct margins measuring 0.5 cm in diameter was noted on the left labiaum minus (Figure 1, arrow). The lesions were not palpable.

There were no lesions on the rest of the vagina or vulva, or on general skin examination. Nevertheless, she had been suffering from urinary incontinence for a considerable period of time. Since the possibility of melanoma could not be ruled out on clinical grounds, three punch biopsies were performed, two from the irregular patch and one from the satellite lesion. The patient received no previous local treatment.

### 2.2. Histopathological Findings

The first punch biopsy from the hyperpigmented patch revealed mild spongiosis and exocytosis of lymphocytes associated with a mild to moderate lichenoid infiltrate in the subepithelial stroma, composed of lymphocytes, plasma cells and neutrophils (Figure 2A). Abundant deposition of the haemosiderin pigment was observed in the same area (Figure 2B). The second punch biopsy, also from hyperpigmented patch, additionally demonstrated numerous extravasated erythrocytes in the subepithelial stroma (Figure 2C). No melanocytic proliferation was identified and S100 protein immunohistochemistry only showed isolated normal melanocytes along the dermal epidermal junction (Figure 2D).

The third punch biopsy, from the satellite lesion, revealed a prominent lichenoid tissue reaction (Figure 3A) associated with focal extravasation of lymphocytes into the epidermis, with basal necrotic keratinocytes (Figure 3B). The lichenoid cell infiltrate consisted of lymphocytes, plasma cells and histiocytes (Figure 3C).

The diagnosis of vulvar lichen planus presenting as chronic vulvar purpura was set based on a review of histopathological samples. After the biopsy, the patient was scheduled for a follow-up in 3 months and received no treatment due to lack of clinical symptoms relating to her condition.

## 3. Discussion

Chronic vulvar purpura is a clinical term used for different chronic inflammatory dermatoses presenting as red bluish or violaceous discolorations on the vulva, often associated with cayenne pepper-like speckling [1]. The most common diseases associated with chronic vulvar purpura include Zoon vulvitis, lichen aureus, pigmented purpuric dermatosis and vulvar lichen planus. While Zoon vulvitis has generally been regarded as a mucosal reaction pattern to various local insults, such as mechanical trauma or irritation, lichen aureus is a localized variant of pigmented purpuric dermatosis, a group of disorders likely to be associated with delayed hypersensitivity reactions or local vascular abnormalities [1,2]. Furthermore, vulvar lichen planus is an immune-mediated reaction of activated T cells to unidentified antigenic stimuli [4]. Importantly, chronic vulvar purpura may mimic melanocytic lesion/proliferation on clinical grounds, necessitating the exclusion of melanoma, as in the current case.

Zoon vulvitis, also designated vulvitis plasmacellularis, presents clinically as a well-demarcated erythematous or ecchymotic patch(s) most commonly localized on the inner aspects of the labia minora, vestibule/introitus, urethral meatus and the clitoris [1]. The histological features depend upon the stage of development, e.g., evolution of the lesion. While early changes can be entirely non-specific with normal or slightly acanthotic epidermis, focal parakeratosis and patchy lichenoid infiltrate, established lesions most often reveal atrophy of the epidermis with superficial erosions, dense band-like infiltrate in the dermis/subepithelial stroma with variable amounts of plasma cells and extravasated erythrocytes [2,5]. In addition, the presence of neutrophils in the upper parts of the epidermis and the so-called crowding of basal keratinocytes has been detected in 80% and 93% of established lesions, respectively [2,6]. More advanced lesions often display deposition of haemosiderin pigment in the upper dermis/subepithelial stroma and variable degrees of fibrosis [2]. Importantly, however, neither of the histological parameters mentioned before, or a combination of thereof, is specific to Zoon vulvitis. Namely, the morphological changes in Zoon vulvitis have generally been attributed to a combination of irritation/moisture and mechanical/friction trauma, and are as such fairly non-specific, representing rather a reaction pattern at mucosal sites than a specific disease process [2]. Since such changes can be superimposed on a more specific disease process, for example psoriasis or lichen planus, the diagnostic histopathological features for an underlying unrelated disease can be obscured, preventing its correct recognition. Essentially, histological features such as vacuolar degeneration of basal keratinocytes, necrotic keratinocytes and psoriasiform epidermal hyperplasia are not encountered in Zoon vulvitis, suggesting an unrelated inflammatory process [2]. Based on this concept, Zoon vulvitis can either develop as an isolated condition or represents an unrelated phenomenon in the background of a more specific inflammatory process [1,2].

Purpuric lesions are characterized by red or purple discoloration caused by bleeding into the skin, whereas pigmented vulvar lesions are characterized by a change in color or pigmentation of the skin, caused by an increase in the production of melanin or the accumulation of other pigments. Pigmented vulvar lesions can be benign or malignant and can be caused by a variety of conditions such as freckles, age spots, or melanoma. They can be dark or light and can appear in various sizes and shapes [7,8].

Pigmented purpuric dermatoses presenting at genital sites are extremely rare [1,3]. They are characterised clinically by a persistent golden-brown patch with purpuric appearance and are distinguished histologically by a combination of red blood cell extravasation in the subepithelial stroma, variable amounts of haemosiderin pigment and predominantly lymphocytic inflammatory cell infiltrate [1,3]. Nevertheless, due to the clinical and morphological overlap with other groups of diseases presenting as chronic vulvar purpura, in particular, Zoon vulvitis and pigmented purpuric dermatosis, these conditions most likely share a common pathogenetic process, including mechanical trauma, irritation or local vascular abnormalities [1,3].

Genital lichen planus can develop as an isolated condition in the absence of cutaneous lesions. Three clinical variants are generally recognized, including erosive, classic/papular and hypertrophic lichen planus [4]. In contrast to cutaneous lichen planus, mucosal lichen planus is often associated with spongiosis. Furthermore, band-like inflammatory cell infiltrate in the subepithelial stroma can be patchy and scant, and plasma cells often represent a significant proportion of the inflammatory cell component, mimicking thereby Zoon vulvitis [4]. Nevertheless, the focal interface tissue reaction associated with basal cell hydropic degeneration often helps in distinguishing the two conditions [2,4]. It must be noted that lichen planus can also present in mouth. About 20% of women with oral lichen planus also develop genital lesions [9]. Both can have negative psychological effect on the patient, as they can interfere with daily activities [10,11,12].

The treatment for vulvar lichen planus can vary depending on the severity and presentation of the condition. Treatment may be topical with the use of corticosteroids, calcineurin inhibitors or oral with the use of corticosteroids, azathioprine and mycophenolate mofetil. Thus, conservative or non-pharmacologic management may be considered in the case of asymptomatic patients as in our case [13].

## 4. Conclusions

Although clinical presentations and the histopathological findings of chronic pigmented vulvar lesions are well described in other case reports, there is still a great overlap of findings among possible diagnoses in our case report. Therefore, physicians must consider all of the differential diagnoses and possible underlying diseases before reaching a conclusion based on clinical and histopathological findings. The final diagnosis that is needed for proper treatment is challenging because of the resemblance and the morphological spectrum of purpuric vulvar diseases. A proper diagnosis can only be made by a healthcare professional after a physical examination and possibly a biopsy.

## Figures and Tables

**Figure 1 medicina-59-00206-f001:**
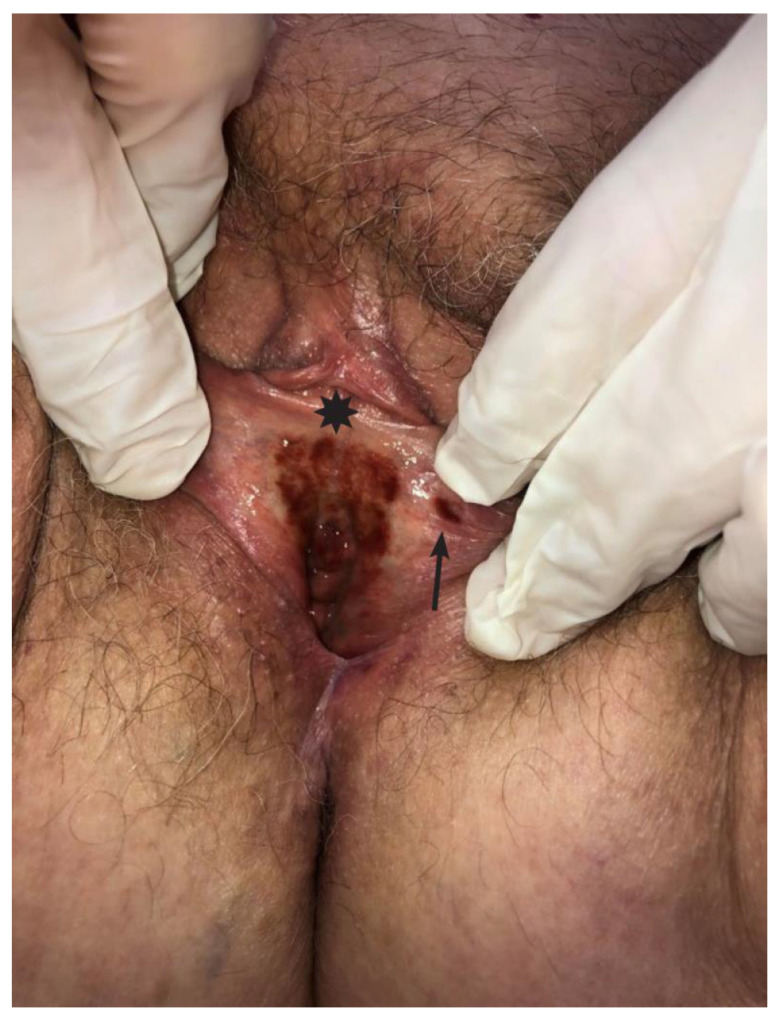
Hyperpigmented patch on the vulva (asterix). Note also a satellite papule (arrow).

**Figure 2 medicina-59-00206-f002:**
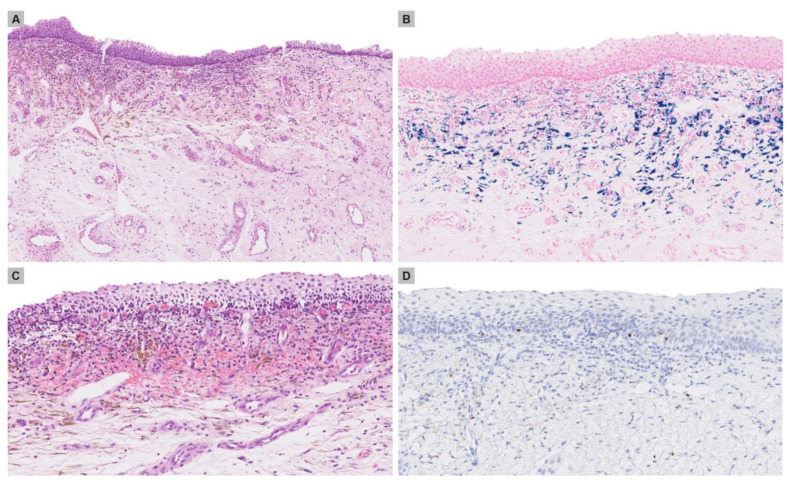
(**A**) Punch biopsy of the hyperpigmented patch on the vulvar revealing mild acanthosis of the epidermis and somewhat bandlike infiltrate in the subepithelial stroma, with an abundance of brownish pigment. (**B**) Perls stain confirming the presence of haemosiderin pigment in the subepithelial stroma. (**C**) In some areas, prominent extravasation of erythrocytes was also seen. (**D**) S100 immunohistochemistry depicting isolated melanocytes along the epidermo-dermal junction. Haematoxylin and eosin, original magnification (**A**) ×40, (**C**) ×60. Perls stain, original magnification (**B**) ×60. S100 immunohistochemistry, original magnification (**D**) ×80.

**Figure 3 medicina-59-00206-f003:**
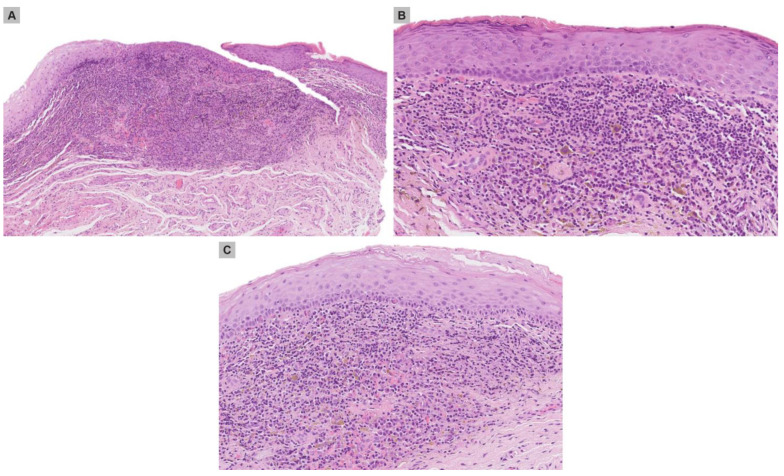
(**A**) Punch biopsy of the papule with atrophic epidermis and moderately dense band-like infiltrate in the subepithelial stroma. (**B**) Mild spongiosis with minimal interface change, including lymphocyte exocytosis, was also present. Note also lichenoid tissue reaction with deposition of haemosiderin pigment. (**C**) Isolated necrotic keratinocytes were seen along the dermal epidermal junction. Haematoxylin and eosin, original magnification (**A**) ×50, (**B**) ×100, (**C**) ×100.

## Data Availability

The data are available upon request.

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
