# Peer review of "Vulvar Lichen Planus Presenting as Chronic Vulvar Purpura"

_medicina, 2023, doi:10.3390/medicina59020206_

Round 1

Reviewer 1 Report

This is a case report regarding vulvar lichen planus presenting with a purpuric patch. The most common form of vulvar lichen planus is the erosive one. However, bright red areas and purpuric clinical lesions are not uncommon in vulvar lichen planus.

I would suggest some changes:

- Histopathological images have a really good quality and are really illustrative. However, as I see the clinical picture, the patch seems to me more like a reddish to brownish patch (more similar to a clinical hemorrhagic suffusion) rather than a greyish to blackish patch, which would suggest a melanocytic lesion. Demoscopy could have been useful to clear this issue up. With this clinical image, I would consider lichen planus and Zoon’s vulvitis as the main differential diagnosis.

- I miss some information in the anamnesis, such as the presence of flares of pruritus, burning or pain, which are typical symptoms of vulvar lichen planus.

- Treatment for this condition should include the treatments used for vulvar lichen planus, rather than those for pigmented purpuric dermatosis, as the diagnosis of this case is lichen planus, while the purpuric images are just the clinical presentation.

- In the conclusion, the authors mention again “pigmented vulvar diseases”, although the clinical presentation is more a purpuric one (as they state in the title), rather than suggesting a melanocytic lesion.

Author Response

We thank the reviewer for his/her valuable insights and constructive comments that have contributed to a better quality of our manuscript. Below are our responses in italics:

This is a case report regarding vulvar lichen planus presenting with a purpuric patch. The most common form of vulvar lichen planus is the erosive one. However, bright red areas and purpuric clinical lesions are not uncommon in vulvar lichen planus.

I would suggest some changes:

  • Histopathological images have a really good quality and are really illustrative. However, as I see the clinical picture, the patch seems to me more like a reddish to brownish patch (more similar to a clinical hemorrhagic suffusion) rather than a greyish to blackish patch, which would suggest a melanocytic lesion. Demoscopy could have been useful to clear this issue up. With this clinical image, I would consider lichen planus and Zoon’s vulvitis as the main differential diagnosis.

We have rephrased/reworded introduction and the first part of clinical presentation of the manuscript to improve its clarity and concision. We believe it is now evident that the Zoon’s vulvitis and lichen planus are the main differential diagnosis.

  • I miss some information in the anamnesis, such as the presence of flares of pruritus, burning or pain, which are typical symptoms of vulvar lichen planus.

We have expanded the clinical presentation section of the manuscript to provide a more complete description of the case. We have explicitly mentioned that the patient did not complain about the presence of flares of pruritus, burning, or pain.

  • Treatment for this condition should include the treatments used for vulvar lichen planus, rather than those for pigmented purpuric dermatosis, as the diagnosis of this case is lichen planus, while the purpuric images are just the clinical presentation.

We appreciate the reviewer's suggestion to include the treatments of vulvar lichen planus instead of those for pigmented purpuric dermatosis. We have now included this information and additional references in the discussion of the manuscript.

  • In the conclusion, the authors mention again “pigmented vulvar diseases”, although the clinical presentation is more a purpuric one (as they state in the title), rather than suggesting a melanocytic lesion.

We have revised the manuscript to address the reviewer's concerns about pigmented vulvar diseases. The revised manuscript now includes additional discussion about the purpuric and pigmented valvular lesions to address this issue.

Reviewer 2 Report

Dear Correspondig Author the article  is well conducted but it is necessary to modify some things 

I suggest you to modify it and add the type of article.

Add recent references about the topic of the article, Preferably a published articles should be with 90 or more references. Useful papers:  [PMID: 29460525]; [DOI: 10.11138/orl/2016.9.2.054] ; [PMID: 29460524]; [DOI: 10.1055/s-0042-1757906]

- review the grammar of your article, 

-Please expand conclusion section with main results and future perspectives of this study

Thank You,

Kind Regards

Author Response

We thank the reviewer for his/her valuable insights and constructive comments that have contributed to a better quality of our manuscript. Below are our responses in italics:

Dear Correspondig Author, the article is well conducted but it is necessary to modify some things I suggest you to modify it and add the type of article.

  • Add recent references about the topic of the article, Preferably a published articles should be with 90 or more references. Useful papers: [PMID: 29460525]; [DOI: 10.11138/orl/2016.9.2.054] ; [PMID: 29460524]; [DOI: 10.1055/s-0042-1757906]

We have added 3 new references to the manuscript to provide a more comprehensive view of the current literature on the topic.

  • review the grammar of your article,

We have made significant grammatical corrections to the manuscript to improve its readability. The article was proof-read by an expert on the field.

  • Please expand conclusion section with main results and future perspectives of this study

We have rephrased/reworded the conclusion of the manuscript to improve its clarity and concision. Moreover, we have emphasized that the chronic pigmented vulvar lesions can have many histopathological presentations which resemble other pathologies. Therefore, rigorous diagnosis is needed for proper treatment.

e other pathologies. Therefore, rigorous diagnosis is needed for proper treatment.

Round 2

Reviewer 1 Report

Regarding the sentence “the lesions were not palpable”, which has been added to the first paragraph of the clinical presentation, the term “plaque” identifies, by definition, an elevated lesion (larger than a papule). I think the authors mean “patch”, which is a flat lesion (larger than a macule). In fact, the lesion of the picture is actually a patch, which is not palpable.

Regarding the paragraph about the treatment of lichen planus, the second sentence is confusing and repetitive. Tacrolimus ointment is a topical calcineurin inhibitor. Maybe the authors want to say “tacrolimus ointment or other calcineurin inhibitors (pimecrolimus)”, although saying only "calcineurin inhibitors" would be appropriate and less reiterative.

Author Response

We thank the reviewer for his/her valuable insights and constructive comments that have contributed to a better quality of our manuscript. Below are our responses in italics:

  • Regarding the sentence “the lesions were not palpable”, which has been added to the first paragraph of the clinical presentation, the term “plaque” identifies, by definition, an elevated lesion (larger than a papule). I think the authors mean “patch”, which is a flat lesion (larger than a macule). In fact, the lesion of the picture is actually a patch, which is not palpable.

We thank the reviewer for pointing out the oversight in the description of the clinical presentation. We have now corrected this error and apologize for any confusion it may have caused. We agree that the term patch is more appropriate than the term plaque.

  • Regarding the paragraph about the treatment of lichen planus, the second sentence is confusing and repetitive. Tacrolimus ointment is a topical calcineurin inhibitor. Maybe the authors want to say “tacrolimus ointment or other calcineurin inhibitors (pimecrolimus)”, although saying only "calcineurin inhibitors" would be appropriate and less reiterative.

We apologize for any confusion caused by the section about the treatment of lichen planus. We have made the necessary changes, and only included the calcineurin inhibitors.